# Relationship between Connective Tissue Morphology and Lower-Limb Stiffness in Endurance Runners. A Prospective Study

**DOI:** 10.3390/ijerph18168453

**Published:** 2021-08-10

**Authors:** Alberto Rubio-Peirotén, Felipe García-Pinillos, Diego Jaén-Carrillo, Antonio Cartón-Llorente, Ferrán Abat, Luis E. Roche-Seruendo

**Affiliations:** 1Campus Universitario, Universidad San Jorge, Villanueva de Gállego, 50830 Zaragoza, Spain; djaen@usj.es (D.J.-C.); acarton@usj.es (A.C.-L.); leroche@usj.es (L.E.R.-S.); 2Department of Physical Education and Sport, Faculty of Sports Sciences, University of Granada, 18071 Granada, Spain; fegarpi@gmail.com; 3Department of Physical Education, Sports and Recreation, Universidad de La Frontera, Temuco 01145, Chile; 4Department of Sports Orthopaedics, ReSport Clinic Barcelona, University of Health and Sports Sciences, Blanquerna—Ramon Llull, 08034 Barcelona, Spain; abat@resportclinic.com

**Keywords:** foot behavior, tendon, stretch-shortening cycle, running

## Abstract

Background: The lower limb behaves like a spring compressing and decompressing during running, where lower-limb stiffness is one of the most influential factors. This prospective observational study is aimed at examining the relationship between the connective tissue morphology and lower-limb stiffness and investigating whether the barefoot/shod condition influences on such relationship. Methods: 14 male amateur runners (10-km time trial <50′) were included. Data were recorded over one session, where participants ran 2 trials (i.e., barefoot and shod conditions) of 3 minutes at 12 km/h, where running spatiotemporal parameters and vertical (Kvert) and leg stiffness (Kleg) were obtained. Prior to testing trials, thickness and cross-sectional area (CSA) were recorded for Achilles (AT) and patellar tendons (PT) and plantar fascia (PF) with ultrasound. Results: Under barefoot condition, a positive correlation was found between Kleg and AT-thickness and CSA and PF-thickness; and between Kvert and AT-thickness and PF thickness. Under shod condition, a positive correlation was found between Kleg and PT-CSA and PT-thickness, and between Kvert and PT-CSA and PT-thickness. Conclusions: The results reveal a specificity of the relationship between the lower-limb stiffness and the morphology of the connective tissue. Greater tendon shows higher lower-limb stiffness when that tendon is specially demanded by the function.

## 1. Introduction

During running, the lower limb behaves like a spring, which compresses and decompresses over the different gait phases [1,2]. In this spring-like model mechanical energy is stored due to the compression provoked by the runner’s body mass during the eccentric phase of stance [3,4]. The release of that energy occurs during the concentric phase of stance facilitating the subsequent movements [3,4]. The behavior of such leg-spring function can be influenced by several factors such as the foot strike pattern (FSP) [5], footwear condition [6], sex difference [7], velocity [8], fatigue [9] or surface type [10]. It has been shown that two of the most important elements of the spring-like behavior of the leg while running are the stretch-shortening cycle (SSC) [11] and the lower-limb stiffness [12].

In both SSC and lower-limb stiffness the muscle-tendon unit is essential. The properties of the muscle-tendon unit play a vital role for the proper functioning of the SSC (i.e., the proper muscle shortening speed allows the optimization of accumulation and release of energy) [13]. Moreover, the relation between tendon and lower-limb stiffness has been previously demonstrated [14]. Rogers et al. [14] showed an association between AT and lower-limb stiffness in terms of vertical (Kvert) and leg stiffness (Kleg). A greater AT stiffness was linked to improved running performance [14].

Previous studies about lower-limb stiffness used Kvert and Kleg as variables to characterize such neuromuscular mechanism [15]. Kvert refers to the resistance of the center of mass to vertical displacement after being subjected to the reaction force of the ground [16], while Kleg was defined as the mechanical behavior of the structural components of the leg (i.e., tendons, joints, muscles) is shown by the change in leg length during eccentric phase [17]. It has been demonstrated that both Kvert and Kleg highly contribute to the spatiotemporal running gait characteristics [18]. Lower-limb stiffness operates according to the specificity principle behind a particular task [19]. Thus, changes in the specific task such as the FSP might alter the behavior of neuromuscular elements (i.e., lower-limb stiffness) in that activity. It is known that the FSP is highly influenced by the shod/barefoot condition [20]. Generally, runners under shod condition, showed a rearfoot strike pattern, and runners under barefoot condition tended to show midfoot or forefoot strike pattern [21]. During running, the FSP determines that some muscles will be especially demanded. In this way, rearfoot strike pattern, under shod condition, shows higher demand of knee extensors [22] and midfoot or forefoot strike pattern, under barefoot condition, especially demands the ankle plantar flexors [23].

As far as the authors know, the relation between morphological characteristics of the main lower-limb connective tissue (i.e., thickness and cross-sectional area (CSA)), and lower-limb stiffness during running remains in debate. Monte et al. [24] found that runners with a greater AT-CSA showed greater Kvert. Similarly, it is unclear whether the shod/barefoot condition, and consequently the FSP, may influence such relationship.

Therefore, the aim of this study is twofold: (i) to examine the likely relation between the morphologic characteristics of the connective tissue and the lower- limb stiffness in terms of Kvert and Kleg, and (ii) to determine whether the shod/barefoot condition influences the mentioned relation between connective tissue and lower-limb stiffness. We hypothesized that higher values in tendon thickness and CSA would be found alongside greater Kvert and Kleg values, especially when the shod/barefoot condition demands the corresponding connective tissue.

## 2. Materials and Methods

### 2.1. Type of Design

Prospective observational study.

### 2.2. Subjects

Fourteen recreationally trained male endurance runners (*n* = 14; age: 27.4 ± 6.3 years; height: 1.75 ± 0.07 m; body mass: 70.9 ± 7.9 kg; BMI: 23.1 ± 2.3) participated in this prospective observational study. All subjects met the inclusion criteria: (i) from 18 to 40 years old, (ii) 3 or more running sessions per week, (iii) 10-km time trial ≤50 min, and (iv) not suffering from any active known injury. Criteria ii, iii and iv refer to the last 6 months before the data collection. After receiving detailed information on the objectives and procedures of the study, each participant signed an informed consent, which complied with the ethical standards of the World Medical Association’s Declaration of Helsinki (2013). It was made clear that the subjects were free to leave the study at any moment. The study was approved by the Institutional Review Board of the San Jorge University (Nº 006-18/19).

### 2.3. Anthropometric Measurements

Body mass (kg) and height (m) were determined using a weighing scale (Tanita BC-601; TANITA Corp., Maeno-Cho, Itabashi-ku, Tokyo, Japan) and a stadiometer (SECA 222; SECA Corp., Hamburg, Germany) for descriptive purposes. The leg length was measured from the great trochanter to the floor in a standing position for lower limb stiffness calculation. 

### 2.4. Tendon Morphology Characteristics

A high-definition ultrasound images were obtained in B-mode with a linear probe 5–16 MHz (LOGIQ S7 EXPERT, General Electric, Germany, 2013). Longitudinal and transversal views of AT, PT and PF were taken before the running protocol. A recent review suggested that US measures of tendon dimensions are reliable, both in terms of relative and absolute reliability [25]. 

To assess the AT, subjects were in prone, with both knees extended and the feet outside of the bed keeping the ankle in neutral position [26,27]. A reference of 3 cm proximal to the insertion of the tendon in the calcaneus bone, measured by the ultrasound device, was used to measure the tendon thickness and CSA [26,27].

The PT was measured with subjects in supine, with both knees bent at 30° [26,27]. A reference of 1 cm distal to the lower pole of the patella, identified by the ultrasound device, was used to assess the tendon thickness and CSA [26,27].

The PF ultrasound assessment was done with subjects in prone position, with both knees in extension, ankles in neutral position and the fingers extended against the surface of the bed [26,27]. A reference, identified by the ultrasound device, located from the anterior edge of the plantar surface of the calcaneus bone vertically to the anterior edge of the PF was used to measure the thickness of the PF [26,27].

For all structures, a frequency of 12 MHz and gain of 100 dB was used. Each measurement was recorded twice by a skilled researcher with more than ten years of experience in diagnostic ultrasound imaging. The selected image was the one considered clearest by the examiner for the subsequent calculation of the morphological variables. Before the statistical analysis, thickness and CSA was measured using the software ImageJ (NIH, Baltimore, MD, USA) [28], using the polygon tool for the CSA.

### 2.5. Procedures

The procedure was performed by every participant under the same conditions and researcher control. Before the start of the testing session, the subjects developed a structured dynamic 5-min warm-up protocol (squatting, lunging, and hinging) [29]. Just after warming up, an accommodation program over 8 minutes [30] was developed by increasing speed by 1 km/h every minute from 8 to 12 km/h. After that, subjects ran under the first footwear condition (shod or barefoot) at a speed of 12 km/h for 3 minutes [31], −6 and −8 strides were analyzed to obtain representative data in healthy adults (95% confidence intervals within 5% of error) [32]. Thereafter, subjects ran under the next footwear condition at 12 km/h for another 3 minutes. The order of the shod or barefoot condition was randomized thus half of the sample started the protocol wearing shoes and the other half running barefoot.

Both running conditions were completed on a motorized treadmill with a slope of 0% (HP cosmos Pulsar 4 P; HP cosmos Sports & Medical, Gmbh, Nußdorf, Germany) and data were recorded for analysis. The completed protocol was illustrated in Figure 1.

### 2.6. Materials and Testing

Data were collected over a 2-trial session in the biomechanics laboratory of the university during March and April 2019.

#### 2.6.1. Running Spatiotemporal Parameters

Contact time (CT; time between touch-down and take-off of the same foot) and flight time (FT; time between take-off of one foot and touch-down of the other) were measured using a photoelectric cell system (Microgate, Bolzano, Italy), which was previously validated for the assessment of running gait spatiotemporal parameters [33]. The 2 bars of photoelectric cell systems were fixed and stabilized at both sides of the treadmill.

#### 2.6.2. FSP

To determine the FSP a high-definition camera (Imaging Source DFK 33 UX174, The Imaging Source Europe GmbH, Eschborn, Germany) was placed 2 meters lateral to the treadmill, level with the running surface. Videos were sampled at 240 frames per second. Using slow motion video playback, the FSP was determined by one researcher with a wide experience in running biomechanics analysis. This method has been proven valid and reliable previously [34].

#### 2.6.3. Lower-Limb Stiffness

The Kvert and the Kleg were measured to determine lower-limb stiffness [15] using the sine-wave method [24,35]. In order to follow the Morin’s method, the collection of information such as body mass, leg length, speed, FT, and CT is required to estimate a runner’s Kvert and Kleg. It was shown that Morin’s method determines accurately Kvert and Kleg for intra and inter-day designs (ICCs = 0.86–0.99) [36]. Reliability of Kvert and Kleg have been previously reported [16,36]. 

### 2.7. Statistical Analysis

Descriptive data are presented as mean and standard deviation (SD). The normality distribution of the data was confirmed by Shapiro-Wilk’s test (*p* > 0.05). To determine the intra-rater reliability of the measures related to the morphology of the connective tissue, intra class correlation coefficients (ICCs) were calculated for each parameter. Additionally, the 95% confidence interval (CI) of the ICC value was provided [37]. To analyze the relationship between the morphology of connective tissue and lower-limb stiffness in endurance runners, a Pearson correlation analysis was conducted for the whole group. The following criteria were adopted to interpret the magnitude of correlations between measurement variables: <0.1 (trivial), 0.1–0.3 (small), 0.3–0.5 (moderate), 0.5–0.7 (large), 0.7–0.9 (very large), and 0.9–1.0 (almost perfect) [38]. A cluster k-means analysis matched the whole group into 2 sub-groups regarding lower-limb stiffness, in terms of Kvert and Kleg, for each running condition (i.e., barefoot vs. shod). An analysis of variance (ANOVA) was conducted between the created sub-groups during each running condition (i.e., barefoot vs. shod) for each dependent variable (i.e., connective tissue morphology). The magnitude of the differences between values was also interpreted using the Cohen’s d effect size (ES) (between-group differences) [39]. Effect sizes are reported as: trivial (<0.19), small (0.2–0.49), medium (0.5–0.79), and large (≥0.8) (Cohen, 1988). All statistical analyses were performed using SPSS software version 25.0 (SPSS Inc., Chicago, IL, USA) and statistical significance was accepted at an alpha level of 0.05.

## 3. Results

The mean values for the AT-CSA were 55.43 ± 10.91 mm^2^ and for the AT-thickness 6.28 ± 0.68 mm. For the PT the mean values of PT-CSA were 99.25 ± 22.07 mm^2^ and for the PT-thickness 3.54 ± 0.51 mm. For PF-thickness, the mean value was 2.84 ± 0.36 mm.

An excellent intra-rater reliability was reported for the measures related to the morphology of the connective tissue (ICC > 0.989, 95% CI: 0.913–0.996).

For the FSP, 79% (*n* = 11) of the subjects showed a rearfoot strike pattern and a 21% (*n* = 3) showed a midfoot or forefoot strike pattern, under shod condition. Under barefoot condition, 86% (*n* = 12) of the subjects showed a midfoot or forefoot strike pattern and a 14% (*n* = 2) showed a rearfoot strike pattern.

The Pearson correlation analysis (Table 1) reported significant relationships (*p* < 0.05) between Kvert_shod with AT-thickness (r = −0.577) and PF-thickness (r = −0.513), and between Kleg_barefoot and PF-thickness (r = 0.516).

The cluster k-means analysis matched the whole group into 2 sub-groups for each running condition (i.e., barefoot vs. shod) regarding lower limb stiffness (Table 2).

Table 3 shows a comparison of connective tissue morphology parameters between those sub-groups (higher stiffness group [HSG] vs. lower stiffness group [LSG]) in both barefoot and shod running conditions. In the barefoot condition, no significant differences in the morphology of connective tissue were found between the HSG and the LSG (*p* ≥ 0.05). In the shod condition, the LSG reported higher values in AT-thickness (*p* = 0.023) with large ES (ES = 2.00), whereas the rest of parameters showed no between-group differences.

## 4. Discussion

This study aimed to examine the relation between the morphologic characteristics of the connective tissue and the lower-limb stiffness in terms of Kvert and Kleg. Additionally, the influence of the shod/barefoot condition was assessed in the aforementioned relationship.

The results of our study showed a positive correlation between AT-CSA and Kleg, and between AT thickness and PF thickness and Kvert and Kleg under barefoot condition. However, under shod condition a positive correlation of small magnitude between the CSA and thickness of the PT and Kvert and Kleg was found. Previous studies had demonstrated higher demand of the AT during barefoot running [23] and higher demand of the PT during shod condition [20,22]. Therefore, in light of these results, the major finding of this study was the specificity of the relationship between the morphologic characteristics of the connective tissue and the function the lower-limb stiffness. We hypothesized that greater tendons, in terms of CSA and thickness, would correlate with a greater lower-limb stiffness. However, it seems to be that this hypothesized correlation only occurs when the corresponding connective tissue is specially requested by the function.

The lower-limb stiffness is determined by the specificity principle of a particular task [19]. A factor that can modify the characteristics of a task such as running is the FSP (i.e., forefoot strike pattern reduces ground contact time during running) [40]. It has been previously demonstrated that the shod/barefoot condition can modify the FSP during running [21]. Muñoz-Jiménez et al. [21] showed how half of 80 trained recreational runners with no experience in barefoot running switched from a habitual rearfoot strike pattern to a midfoot or forefoot strike pattern when they were asked to run under barefoot condition. During barefoot running there is a reduction of ground contact time due to a lower plantar flexion movement of the ankle [41]. This way, the AT impulse and the AT loading rates are higher during barefoot running [23]. These findings suggest that barefoot condition specially demands the AT, which might explain that in the present study higher lower-limb stiffness values were found associated with higher values of AT thickness only during barefoot condition. Since, it is under this condition when there is a greater functional demand of the AT, and therefore when higher AT would show higher lower limb stiffness. Similar to authors findings, Monte et al. [24] found that in half-marathon runners, a higher Kvert was associated with a higher AT-CSA in faster runners. In our study both AT thickness and AT-CSA were assessed showing a positive correlation with Kleg, and between Kvert and AT thickness, but only under barefoot condition. However, in both AT tendon values (i.e., CSA and thickness) a change in the correlation with the lower-limb stiffness from shod to barefoot condition was identified. As far as the authors know, that study [24] did not control the FSP of the runners. It would be worth knowing if those faster runners exhibited forefoot strike patterns. If so, the results of Monte et al. [24] would support our findings as those runners who would functionally demand more from their AT are the ones who would show that the greater AT, the greater the lower-limb stiffness.

As mentioned above, a positive correlation was found between the PT thickness and CSA and the Kvert and Kleg under shod condition. Again, these findings could be explained by the specificity between the morphological characteristics of the connective tissue and the lower limb stiffness found in this study. Our findings show that during shod running most of the runners show a rearfoot strike pattern, which is supported by previous research [21]. Runners with a rearfoot strike pattern have shown a higher flexion of the knee during the mid-stance phase [20] that causes a higher knee joint load compared to runners with a forefoot strike pattern [22]. As shown in the introduction section, the lower limb behaves like a spring during running [1,2]. The fact that the rearfoot strike pattern during shod running involves a higher knee flexion movement can facilitate the SCC in the PT and therefore a higher demand of this tendon. With these results, similar to what occurred with the AT, it might be stated that PTs with a higher thickness and CSA show higher levels of both Kvert and Kleg at shod running.

After compare the groups regarding the lower-limb stiffness, the results of this analysis follow the same direction that discussed above. On one hand, for the AT and PF the HSG shows higher levels of thickness for both structures under barefoot condition, when the AT is functionally more demanded. On the other hand, for the PT the HSG shows higher levels of thickness and CSA under shod condition, when that tendon is functionally more demanded. Despite the specificity of the relation between connective tissue and lower-limb stiffness, some values were not significant. These results might be explained in regards with the great variability of factors which affect lower-limb stiffness. Variables such as sex difference [7], fatigue [9], and surface type [10] have previously demonstrated to influence this neuromuscular mechanism. Consequently, explaining the lower-limb stiffness exclusively based on connective tissue characteristics would be inaccurate. It is necessary to take into account all the aforementioned factors to analyze the lower-limb stiffness within the context of the spring mass model.

Despite the findings reported here, there are some limitations to be considered. The main limitation was the size of the sample. We assessed fourteen male runners capable of completing a 10 km run under 50 min obtaining thus a homogeneous sample. At the same time, that criteria provoked a reduced sample size what could explain why the power of the correlations between lower-limb stiffness and connective tissue were not very large. Ultimately, participants wore their own running shoes improving, therefore, the ecology of the study. Notwithstanding those limitations, the current study examined the relationship between the connective tissue morphologic characteristics and the lower-limb stiffness in endurance runners.

## 5. Conclusions

On the one hand, the present study reveals that, greater AT and PF show higher lower-limb stiffness values under barefoot condition. On the other hand, a greater PT shows higher lower-limb stiffness under shod condition. In the light of these results, the specificity of the task determines the relation between the morphologic characteristics of the connective tissue and the lower-limb stiffness during running. Thereby, from a practical standpoint, the morphologic characteristics of the connective tissue (i.e., CSA and thickness) might be more determinant for the runner’s lower-limb stiffness and, therefore for running performance, when this tissue is specially demanded by the corresponding foot strike pattern. This specificity shown between the structure (i.e., connective tissue) and the function (i.e., lower-limb stiffness) suggests that runners lead to a specific foot strike pattern, without an adapted corresponding tendon, can lead to inadequate performance and possible injuries due to overstress. Therefore, when considering a change in footwear or even running technique expecting a change in the FSP would need an adaptation of the most functionally demanded tendon for this type of foot strike pattern.

## Figures and Tables

**Figure 1 ijerph-18-08453-f001:**
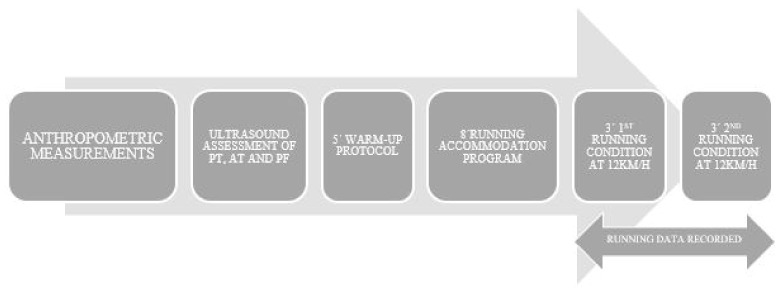
Temporal sequence of the protocol followed in the study.

**Table 1 ijerph-18-08453-t001:** Relationship (Pearson coefficient) between connective tissue morphology and lower-limb stiffness.

	Kvert Shod	Kleg Shod	Kvert Barefoot	Kleg Barefoot
PT-thickness	0.276	0.194	−0.250	−0.470
PT-CSA	0.234	0.125	−0.201	−0.379
AT- thickness	−0.577 *	−0.421	0.041	0.495
AT-CSA	−0.306	−0.311	−0.202	0.051
PF-thickness	−0.513 *	−0.395	0.206	0.516 *

* *p* < 0.05; AT: Achilles’ tendon; CSA: cross-sectional area; Kleg: leg stiffness; Kvert: vertical stiffness; PT: patellar tendon; PF: plantar fascia.

**Table 2 ijerph-18-08453-t002:** Leg and Vertical lower-limb stiffness in barefoot and shod conditions (ANOVA).

	Barefoot		*p*-Value	ES (d)	Shod		*p*-Value	ES (d)
	HSG (*n* = 6)	LSG (*n* = 8)			HSG (*n* = 5)	LSG (*n* = 9)		
Kvert (kN/m)	31.05 (3.21)	26.30 (2.41)	0.05	1.85	36.50 (2.26)	22.99 (2.27)	0.001	6.44
Kleg (kN/m)	14.72 (1.92)	10.48 (1.37)	0.001	2.82	10.34 (0.83)	8.10 (0.77)	0.003	3.04

HSG: higher-stiffness group; Kleg: leg stiffness; Kvert: vertical stiffness; LSG: lower-stiffness group; ES: effect size.

**Table 3 ijerph-18-08453-t003:** Connective tissue morphology regarding lower-limb stiffness groups in both barefoot and shod conditions (ANOVA).

	Barefoot		*p*-Value	ES (d)	Shod		*p*-Value	ES (d)
	HSG (*n* = 6)	LSG (*n* = 8)			HSG (*n* = 5)	LSG (*n* = 9)		
PT-thickness (mm)	3.08 (0.66)	3.55 (0.56)	0.237	0.84	3.64 (0.40)	3.42 (0.62)	0.636	0.44
PT-CSA (mm^2^)	66.68 (6.72)	92.51 (20.89)	0.062	1.68	90.27 (18.43)	86.43 (25.68)	0.826	0.18
AT- thickness (mm)	5.28 (0.40)	4.93 (0.65)	0.404	0.68	4.15 (0.39)	5.15 (0.52)	0.023	2.29
AT-CSA (mm^2^)	50.05 (5.95)	51.95 (7.46)	0.693	0.29	45.21 (1.69)	52.60 (5.02)	0.175	2.00
PF-thickness (mm)	3.11 (0.32)	2.73 (0.41)	0.172	1.09	2.37 (0.16)	2.89 (0.40)	0.103	1.74

AT: Achilles’ tendon; CSA: cross-sectional area; HSG: higher-stiffness group; Kleg: leg stiffness; Kvert: vertical stiffness; LSG: lower-stiffness group; PT: patellar tendon; PF: plantar fascia; ES: effect size.

## Data Availability

The data presented in this study are available on request from the corresponding author. The data are not publicly available due to authors preferences.

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
