# Peer review of "Relationship between Connective Tissue Morphology and Lower-Limb Stiffness in Endurance Runners. A Prospective Study"

_ijerph, 2021, doi:10.3390/ijerph18168453_

Round 1

Reviewer 1 Report

The aims of the study are to assess the relationship between the morphological characteristics of connective tissue and lower-limb stiffness, and to determine whether the shod/barefoot condition influences this relationship. The paper is well structured, but there are some aspects that should be improved:

The introduction is complete, presenting the most relevant aspects.

In material and methods: a complete description is given although some aspects are not clear:

In line 112 it is explained that the measurements are performed twice by a single experienced researcher, but it is not specified which measure is finally used.

The results are presented clearly and concisely in the section on results, but there are statistical tests cited in the Material and Methods that are not referred in the results, specifically the Cohen's d effect size. The magnitude of the correlations is not specified in the text nor in the tables (Table 1). And there are not any reference to descriptive data on tendon measurements.

The presentation of the tables needs to include information on the data included in the tables.

The discussion is well structured, although line 216 should include that the positive correlation found is small in magnitude.

The conclusions are concise and based on the results, but the results should be interpreted with caution.

Author Response

We very much appreciate your constructive comments, useful information and your time. Thanks to this review, our manuscript was substantially improved. Responses to your comments are highlighted in yellow.

The aims of the study are to assess the relationship between the morphological characteristics of connective tissue and lower-limb stiffness, and to determine whether the shod/barefoot condition influences this relationship. The paper is well structured, but there are some aspects that should be improved:

The introduction is complete, presenting the most relevant aspects.

Thanks.

In material and methods: a complete description is given although some aspects are not clear:

In line 112 it is explained that the measurements are performed twice by a single experienced researcher, but it is not specified which measure is finally used.

The selected measurement was the one that showed a clearer image for the subsequent calculation of the morphological variables. This information has been added to the manuscript (lines 114-115).

The results are presented clearly and concisely in the section on results, but there are statistical tests cited in the Material and Methods that are not referred in the results, specifically the Cohen's d effect size. The magnitude of the correlations is not specified in the text nor in the tables (Table 1). And there are not any reference to descriptive data on tendon measurements.

Cohen's d effect size added in tables 2 and 3. Descriptive data on tendon measurements added (Lines 155-157)

The presentation of the tables needs to include information on the data included in the tables.

Sorry but we do not understand what the reviewer is referring to with this comment. The titles of the tables explain their contents. Can you specify what information should be included? Thanks.

The discussion is well structured, although line 216 should include that the positive correlation found is small in magnitude.

Done

The conclusions are concise and based on the results, but the results should be interpreted with caution.

Thanks for the appreciation, we will take it into account.

Reviewer 2 Report

Dear authors:

It has been a pleasure to review your paper about “Relationship between connective tissue morphology and lower-limb stiffness in endurance runners. A prospective Study ” it’s necessary to change it. You can see below the recommendation

In section method

Can you include a subheading type of design?

In section results

Can you include the measures of the variables?

I can´t see the Cohen’s d effect size, can you include these dates?

Author Response

We very much appreciate your constructive comments, useful information and your time. Thanks to this review, our manuscript was substantially improved. Responses to your comments are highlighted in yellow.

It has been a pleasure to review your paper about “Relationship between connective tissue morphology and lower-limb stiffness in endurance runners. A prospective Study ” it’s necessary to change it. You can see below the recommendation

In section method

 Can you include a subheading type of design?

 Done. Lines 76-77

 In section results

 Can you include the measures of the variables?

Done. Lines 185-187

 I can´t see the Cohen’s d effect size, can you include these dates?

Done. Tables 2 and 3.

Reviewer 3 Report

1. The sample is very small (N=14), has an a priori power analysis been
conducted? Are estimations reliable with such a small sample? How can
authors argue that specificity of relationship patterns are not simply
by chance in this small sample?

2. The manuscript contains many abbreviations. Please include a glossary
at the beginning or end of the manuscript explaining all abbreviations.

3. What about other key factors such as changes in e.g. motivation,
endurance, strengths, during the session? Was the barefoot condition
always during the first trial and the shod condition during the second
trial of running? This would be a major limitation since the
aforementioned effects of time may be responsible for the differences
observed between the two conditions. Ideally one needs to counterbalance
the order of conditions (i.e., half of participants starting with
barefoot and the other half of participants starting with shod).

Author Response

We very much appreciate your constructive comments, useful information and your time. Thanks to this review, our manuscript was substantially improved. Responses to your comments are highlighted in yellow.

  1. The sample is very small (N=14), has an a priori power analysis been

conducted? Are estimations reliable with such a small sample? How can

authors argue that specificity of relationship patterns are not simply

by chance in this small sample?

The N used is for convenience and due to the complexity of adding ultrasound measurements. This N is similar to that used in studies with similar characteristics such as those of Farris et al 2012 (n=12), Neves et al 2014 (n=20), Magnusson et al 2003 (n=12), Rosager et al 2002 (n=10). The fact of using a sample only of male subjects in order to homogenize the sample is a factor that decreases its size, as discussed in the limitations.

  1. The manuscript contains many abbreviations. Please include a glossary at the beginning or end of the manuscript explaining all abbreviations.

Done. Lines 318-333

  1. What about other key factors such as changes in e.g. motivation,

endurance, strengths, during the session? Was the barefoot condition

always during the first trial and the shod condition during the second

trial of running? This would be a major limitation since the

aforementioned effects of time may be responsible for the differences

observed between the two conditions. Ideally one needs to counterbalance

the order of conditions (i.e., half of participants starting with

barefoot and the other half of participants starting with shod).

The present study assesses the relationship between morphological characteristics of the connective tissue and stiffness of the lower limb, but we appreciate the variables that the reviewer comments and we will take them into account for future studies.

The manuscript indicates that the order of the shod / barefoot condition was randomized (lines 130-131). We add that half of the sample started the protocol in footwear condition and the other half barefoot (lines 131-132).

Round 2

Reviewer 1 Report

Thanks to the authors for taking my suggestions into account.

In relation to tables, they should indicate the statistical test they represent and the statistical data presented.

Author Response

We very much appreciate your constructive comments, useful information and your time. Thanks to this review, our manuscript was substantially improved. Responses to your comments are highlighted in yellow.

In relation to tables, they should indicate the statistical test they represent and the statistical data presented.

Done. Tables 2 and 3.

Reviewer 3 Report

I don't have further considerations about this paper. 

Author Response

Thanks.